# STExplainer: Global Explainability of GNNs via Frequent SubTree Mining

## Abstract

The need for transparency and interpretability in critical domains has led to an increasing interest in understanding the inner workings of Graph Neural Networks (GNNs). While local-level GNN explainability has been extensively studied to find important features within individual graph samples, recent research has emphasized the importance of global explainability of GNNs by uncovering global graphical concepts in a dataset underlying GNN behaviors. In this paper, we look into the intrinsic message-passing mechanism of standard GNNs and introduce a new method, STExplainer, to directly extract global explanations of GNNs using rooted subtrees on a dataset level instead of per instance. Unlike existing global explainers, which typically identify clusters of instance-level explanations or aggregate local graphical patterns into prototypes represented as latent vectors or rely on human-defined natural language rules, our approach extracts more intuitive global explanations through rooted subtree patterns and subgraph patterns, along with their associated relative importance scores, without relying on any instance-level explainers. We empirically demonstrate the effectiveness of our approach in extracting meaningful and high-quality global explanations on both synthetic and real-world datasets. The global explanations extracted by STExplainer are faithful to the original GNNs and distinguishable among different classes.

## 1 Introduction

Graph Neural Networks (GNNs) (Kipf & Welling, 2017; Xu et al., 2019; Hamilton et al., 2017) are recognized for their remarkable scalability and expressive power in solving graph-related tasks, including node and graph classifications. However, the black-boxed nature of GNN predictions poses limitations to their utilization in critical domains, e.g., healthcare, finance, and autonomous systems, where decisions made by a model can have significant real-world impacts. This has motivated the research on GNN explainability, which is crucial to ensuring the transparency and trustworthiness in GNN decision making.

Existing works on GNN explainability can be categorized mainly into local-level explainability (Ying et al., 2019; Luo et al., 2020; Vu & Thai, 2020; Yuan et al., 2021; Shan et al., 2021; Bajaj et al., 2021; Lin et al., 2021; Wang et al., 2021; Feng et al., 2022; Xie et al., 2022; Zhang et al., 2022) and global-level explainability (Magister et al., 2021; Azzolin et al., 2023; Xuanyuan et al., 2023). While local-level explainability focuses on identifying important nodes, edges or subgraphs behind a GNN model's specific predictions, these explanations are generated per individual data instance. In contrast, recent work emphasizes global explainability which aims to provide a more comprehensive understanding of a model's behavior on a given dataset, extending its scope beyond predictions on individual data instances. By distilling the model's behaviour into coherent global patterns, global explainers aim to capture the overarching graphical patterns or concepts across the entire dataset underlying the decisions made by a GNN regarding a class of samples, and thus offer a systematic view of the model's overall functioning. This broader perspective also plays a pivotal role in debugging and continually improving GNNs.

In this paper, we propose a new approach to global-level GNN explanation, named **SubTree Explainer (*STExplainer*)**, that extracts subgraph patterns critical to the decision making toward a class of graphs by mining important rooted subtrees, which align more closely with the message-passing

mechanism of GNNs, over the entire dataset. The overview of the approach is illustrated in Figure 1. Our contributions can be summarized as follows:

i) Unlike prior methods that cluster and aggregate concepts found by a local explainer, our method directly enumerates and extracts $L$-hop *rooted subtrees* across the entire dataset based on frequency (which is tunable to adjust coverage and efficiency), where $L$ is the number of layers in the GNN, without relying on a local explainer as a prior step.

ii) Unlike existing global explainers, which use prototypes of local explaining concepts represented in a latent encoding space as explanations (Magister et al., 2021; Azzolin et al., 2023), or rely on human-defined natural language rules (Xuanyuan et al., 2023), we generate induced explanation subgraphs on the dataset level from all the subtrees mined. Since STExplainer produces subgraph-based concepts rather than latent representations of prototypes or manually defined rules, STExplainer can offer more intuitive and visualizable interpretation of GNN behavior on each class of graph samples.

iii) We provide an optimization method to quantitatively compute the relative importance of the extracted rooted subtrees and their induced subgraphs, identifying how significant or relevant the subtree or subgraph patterns are in predicting the target class.

iv) Our approach further utilizes node embeddings to represent graphical patterns of subtrees, and thus offers an efficient way to quickly assess whether the knowledge gained (subgraph concepts) from a training set is present in and relevant to decision making for any new graph instance.

Through extensive evaluations, we demonstrate the advantages of our proposed approach in providing high-quality, meaningful and intuitive global explanations for different classes of samples on various synthetic datasets and real-world datasets, in comparison to existing methods on global GNN explainability. We also show that the global concepts found by STExplainer are faithful to GNN predictions and class-distinguishing, while covering a majority of data samples in the datasets.

## 2 RELATED WORK

Global-level GNN explainability is a relatively nascent research direction, with limited exploration and investigation. GCExplainer (Magister et al., 2021) adapts the well-known Automated Concept-based Explanation (ACE) (Ghorbani et al., 2019) approach to GNNs by introducing human expertise into the process. It employs the $k$-Means clustering algorithm on GNN-generated embeddings to group nodes or graphs of similar types, with each cluster serving as a representation of a concept. Subsequently, human experts engage in the analysis of instances within each cluster to discern clear rules or subgraph patterns that represent these concepts. While this method is conceptually sound, it faces practical challenges in real-world applications due to reliance on human experts.

Rather than directly employing node or graph embeddings, GLGExplainer (Azzolin et al., 2023) leverages local subgraph explanations generated by PGExplainer (Luo et al., 2020). These subgraph explanations are fed into the original GNN to produce subgraph embeddings. Following this, GLGExplainer applies prototype learning to these subgraph embeddings, forming clusters and identifying a prototype within each cluster. These prototypes are essentially vectors in the latent space, initially randomized from a uniform distribution, and learned alongside other architectural parameters. These prototypes are subsequently employed in an E-LEN model to derive a Boolean formula that mimics the behavior of the underlying GNN. As a result, the global explanation provided by this method is represented in the form of a Boolean formula, where the premises are the latent prototypes.

GCNeuron (Xuanyuan et al., 2023) adopts a different approach by formulating human-defined rule templates in natural language and considering graphs with certain nodes masked as concepts. When a natural language rule can describe a masked graph, the masked graph is identified as an interpretable concept. Subsequently, they follow Mu & Andreas (2020) to perform beam search over the space of compositional concepts for logical compositional rules. The highest-scoring compositional concepts that enhance the prediction probability of the target class are then identified as the global explanations.

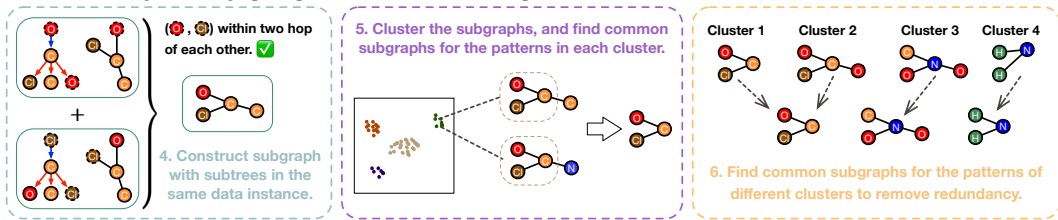

Figure 1: Overview of STExplainer. (a) In this phase, we extract subtree patterns as the explanations. Step 1: We enumerate all the subtrees in the dataset and count their frequency. Then, we pick only top $T$ subtrees with the highest frequencies as candidates. We use the embeddings of the rooted nodes to represent candidate subtree embeddings. Step 2: We feed the embeddings of $T$ subtrees to an MLP. The output vector is then used to aggregate the subtrees. The weighted embedding is fed to GNN classifier to optimize the prediction at the target class. Step 3: Remove unimportant subtrees. (b) In this phase, we obtain global subgraph explanations based on subtrees. Step 4: We construct the induced subgraph explanation from the subtrees in each data instance. Step 5: We cluster all the subgraphs obtained from different instances and find the common pattern shared by all subgraphs in each cluster. Step 6: Find common patterns across different clusters to remove redundancy.

Generation-based explainers (Yuan et al., 2020; Wang & Shen, 2022) train a graph generator or generative models to produce numerous new data samples for the target class, which do not provide clear concepts, leaving the human observer to draw a conclusion. Global counterfactual explainer (Huang et al., 2023) greedily finds graph edits that change the prediction labels, which provide insights from the perspective of counterfactual reasoning, whereas we focus on factual concepts in our work.

## 3 BACKGROUND

### 3.1 GRAPH NEURAL NETWORKS

Let $\mathcal{G} = (\mathcal{V}, \mathcal{E})$ be a graph with the associated nodes set $\mathcal{V}$, edges set $\mathcal{E}$, and $N = |\mathcal{V}|$ represents the number of nodes. A GNN model $f(\mathbf{X}, \mathbf{A})$ maps the node features $\mathbf{X} \in \mathbb{R}^{N \times d}$ of dimension $d$ and the adjacency matrix $\mathbf{A} \in \mathbb{R}^{N \times N}$ indicating the existence or absence of edges $\mathcal{E}$ to a target output, such as node labels, graph labels, or edge labels. Let $l$ be a message-passing layer in the GNN. At layer $l$, the GNN aggregates the neighbourhood information for each node $v \in \mathcal{V}$ with the representation $\boldsymbol{h}_v^{(l-1)}$, and embeds the information into the next layer representation $\boldsymbol{h}_v^{(l)}$. Typical GNNs (Kipf & Welling, 2017; Xu et al., 2019; Hamilton et al., 2017) aggregate the information from the 1-hop neighbours $\mathcal{N}$ of $v$ as

$$\boldsymbol{h}_v^{(l)} = \text{UPDATE}^{(l)} \left( \boldsymbol{h}_v^{(l-1)}, \text{AGG}^{(l)} \left( \left\{ \boldsymbol{h}_u^{(l-1)} : u \in \mathcal{N}(v) \right\} \right) \right), \tag{1}$$

where $\text{UPDATE}^{(l)}$ and $\text{AGG}^{(l)}$ represent the updating and aggregation functions.

### 3.2 SUBTREE PATTERNS

A (rooted) subtree typically refers to a subgraph of a graph, devoid of cycles, and structured like a tree with a designated root node. A subtree of $\mathcal{G}$ can thus be seen as a connected subset of distinct

nodes of G with an underlying tree structure. A $k$-hop subtree is the underlying tree structure within $k$-hop distance from the root node. Similar to the notion of walk that extends the notion of path by allowing nodes to be equal, the notion of subtrees can be extended to subtree patterns (Shervashidze et al., 2011), which can have nodes that are equal (see Figure 1). These repetitions of the same node are then treated as distinct nodes, such that the pattern is still a cycle-free tree. Note that similar to Shervashidze et al. (2011), Xu et al. (2019), and Zhang & Li (2021), the "subtrees" considered in this paper refers to the subtree *patterns*, not strict subtrees. Major GNN variants like GCN and GIN, which are based on the Weisfeiler-Lehman (1-WL) test (Leman & Weisfeiler, 1968), aggregate information from subtree patterns in their message passing layers. This is precisely the reason we look into subtrees.

## 4 METHOD

In this section, we first introduce our methodology for extracting global subtree explanations and then discuss how we extend this to generate global subgraph explanations using the clustering algorithm. The code is uploaded with the supplementary material of our submission.

### 4.1 EXTRACT SUBTREE PATTERNS AS THE GLOBAL EXPLANATIONS

**Collect candidate subtrees based on frequency.** As illustrated in Figure 1 Step 1, to mitigate the influence of noisy data and improve the quality of our candidate explanations, we focus solely on the patterns that occur frequently in the dataset. Traditional Frequent Subtree Mining problem (Chi et al., 2004) is defined as: Given a threshold *minfreq*, for non-isomorphic subtrees $\mathcal{P}$

$$\forall P \in \mathcal{P} : \text{freq}(P, \mathcal{D}) = \sum_{G \in \mathcal{D}} \psi(P, G) \geq \text{minfreq}, \tag{2}$$

where $\mathcal{D}$ is the set of induced trees from the graph instances in the dataset, $\psi$ is a function indicating the frequency of subtree $P$ in the induced tree $G$ of the corresponding data instance. Following this problem setup, we are able to ignore the subtree patterns that barely appear in the dataset.

In our paper, we want to additionally ignore the subtrees shared between the target class and other classes. In other words, let $\mathcal{C}$ be the set of induced trees of the target class, $\mathcal{H}$ be the set of induced trees of other classes, we define the frequent subtrees as

$$\forall P \in \mathcal{P} : \text{freq}(P, \mathcal{D}) = \left| \sum_{G \in \mathcal{C}} \psi(P, G) - \sum_{G \in \mathcal{H}} \psi(P, G) \right| \geq \text{minfreq}. \tag{3}$$

We quantify the magnitude of the frequency difference, as a negative difference indicates the corresponding subtree $P$ appears more often in other classes, which, in global view, could have a negative impact on the prediction of the target class. Hence, we consider these subtrees in our analysis. We collect a set of $T$ candidate subtrees, where the threshold *minfreq* is determined by the minimum frequency of the top $T$ subtrees.

This approach is efficient because, in the case of an $L$-layer Graph Neural Network (GNN), we limit our focus to $L$-hop subtrees within the data samples. So, if each data sample has an average of $\overline{N}$ nodes, and the dataset contains $|\mathcal{G}|$ samples, we only need to enumerate $\overline{N}|\mathcal{G}|$ subtrees. In contrast, when dealing with subgraphs, the enumeration process becomes much more computationally demanding as we would potentially have to consider up to $N!$ subgraphs within each data sample. Therefore, focusing on $L$-hop subtrees in a $L$-layer Graph Neural Network (GNN) is computationally efficient, reducing the enumeration complexity to $\overline{N}|\mathcal{G}|$ compared to the potentially factorial complexity of subgraph enumeration.

**Importance learning for candidate subtrees.** As illustrated in Figure 1 Step 2, after we obtain the top $T$ subtrees in Step 1, we use the corresponding $L$-layer node embeddings to represent the $L$-hop rooted subtrees, and feed the embeddings to a multilayer perceptron (MLP) that outputs a vector $W \in \mathbb{R}^{1 \times T}$. We will use this vector later in Step 3 to represent the importance of subtrees. Next, we perform matrix multiplication between $W$ and the embeddings $H$ of the top $T$ subtrees to obtain the weighted embedding of the dataset. Then, the weighted embedding is passed to the classifier $\phi(\cdot)$ of the original GNN, resulting in the final prediction values at each output class before any softmax

layers. It is worth noting that the parameters in the classifier is fixed throughout the training process of our model.

The training objective is to minimize the prediction loss on the target class that we aim to explain. Additionally, we use a penalty term to limit the total weights that the explanations can take. Formally, the loss function is defined as:

$$\mathcal{L} = -\log \frac{\exp(p_t)}{\sum_{c=1}^{C} \exp(p_c)} + \lambda \|W\|_2 \,, \tag{4}$$

where $p_c$ is the output of $\phi(WH)$ at class $c$, $C$ is the number of classes, $t$ is the index of the target class, and $\lambda$ is a weighing factor.

We incorporate a penalty term for the following reason. As previously mentioned, $WH$ represents the weighted sum of candidate subtree embeddings. By controlling the overall weights, our objective is to encourage the "significant" subtrees to occupy only a minor portion of the "dataset embedding", which can mimic the readout functions typically employed in GNNs. In order words, we introduce this penalty term based on the intuition that GNNs can effectively represent the dataset using a limited number of "significant" subtrees, with the remaining "non-significant" subtrees exerting minimal influence on the prediction of the target class.

For example, in graph classification tasks, the mean-pooling function is typically utilized to obtain the graph embedding for each data instance, which is defined as: $\boldsymbol{r} = \frac{1}{N} \sum_{n=1}^{N} \boldsymbol{h}_n^{(L)}$, where $\boldsymbol{h}_n^{(L)}$ is the last layer node embedding for the $n$-th node. We can observe from this equation that all the subtrees are equally weighted when evaluating the graph embedding, where each subtree takes only a small portion. By limiting the total weights that the candidate subtrees can take, we effectively suppress the influence of non-significant subtrees on predictions, even when they receive greater weights. Conversely, significant subtrees retain the capacity to exert a crucial impact on predictions. Consequently, the model is incentivized to assign greater importance to these significant subtrees during the learning process.

**Global subtree explanations.** Lastly, as illustrated in Figure 1 Step 3, we standardize the acquired weights relative to the one with the maximum magnitude. Then, we select $M$ subtrees, considering their relative importance magnitude, to serve as the global subtree explanations. This approach allows to consider both the positively significant patterns as well as the negatively significant patterns, which is useful for analysing the predictions of a biased class. For example, if there are three classes in a task, the first and the second classes have patterns of $A$ and $B$ respectively, while the third class does not have any outstanding patterns. Then, the explanation for the third class can be: *If neither pattern A nor pattern B is present in the graph, it should be classified into the third class.* In our case, the explanation would be the two patterns $A$ and $B$, both with negative scores.

**Utilize learned global subtree patterns to explain new instances.** The design nature of our subtree explainer provides a straightforward approach to effectively explaining new data samples using the learned concepts. By representing subtree embeddings directly using the embeddings of their root nodes, when a new instance is introduced, we can readily acquire all the subtree embeddings by feeding the new data into the GNN. Subsequently, we cross-reference these subtree embeddings with the learned concepts. If any of these subtrees are found within the learned concepts, it signifies that we can utilize the acquired knowledge to provide explanations for the new data instances.

## 4.2 EXTRACT SUBGRAPH PATTERNS AS THE GLOBAL EXPLANATIONS

Subgraph explanations are able to capture patterns that are larger or smaller than the $L$-hop subtrees, thus offering a more comprehensive insight of the GNNs. However, when we intend to employ these acquired subgraph patterns as global concepts for inference or reasoning on new data instances, mirroring human reasoning, a challenge arises. The obstacle lies in the computational cost associated with enumerating all potential subgraphs within a new graph and comparing them with the learned concepts. In the following, we elaborate on how we address this challenge by generating subgraph explanations with the subtrees previously extracted.

**Unions of subtrees in the same data instance.** In order to cover the frequent patterns that are larger than $L$-hop subtrees, we look into the overlapped subtree explanations in each data instance by examining the $L$-hop nodes of the rooted nodes.

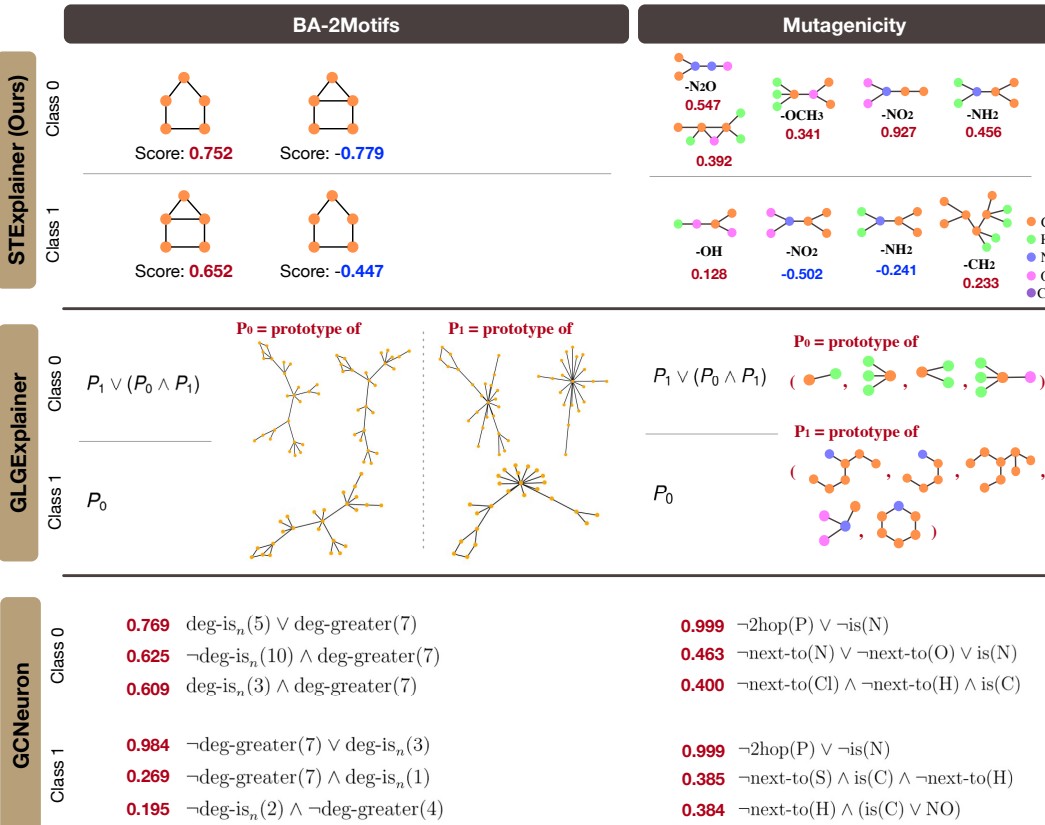

Figure 2: Global explanations by STExplainer (ours), GCNeuron and GLGExplainer. We run both baseline methods so that they explain the same GNN models as our approach. Due to space limit, the explanations on BAMultiShapes and NCI1 datasets are moved to the appendix.

**Definition 1 (Overlapped Subtrees)** *If the rooted node of one rooted subtree is located within the L-hop distance of the rooted node of another rooted subtree, we say that these two rooted subtrees are overlapped subtrees.*

We consider $L$-hop neighbors because, in cases where two rooted nodes are $L$-hop neighbors of one another, certain graph patterns are inherently overlapped. This is due to the fact that the path connecting these two nodes is, by definition, overlapping. When two subtree explanations overlap, they are more likely to function as a cohesive group rather than independently. Therefore, we construct subgraph patterns using the overlapping subtrees. The embedding of a subgraph representing the union of subtrees is acquired by averaging the embeddings of the subtrees in this union. Similarly, the importance score of this union is determined by averaging the importance scores of the subtrees within this union. Additionally, we memorize the embeddings of the subtrees consisting this subgraph for quick assessment on new instances.

**Intersections of the subgraph patterns in the same cluster.** After deriving the subgraph patterns by combining overlapping subtrees within each data instance, we employ the $k$-Means clustering algorithm on these subgraphs. The choice of $k$ is determined based on the mean distance between the subgraph embeddings and the centroid of each cluster. We increase $k$ until the mean distance of all clusters falls below a predefined threshold, denoted as $\tau$. Subsequently, we randomly select $S$ subgraph patterns from each cluster and perform subgraph matching on them, ultimately yielding the intersection of these $S$ subgraphs. This intersection serves as the representative subgraph concept for a given cluster. In cases no common subgraphs are found in a cluster, we simply discard this cluster. This is unlikely since we have constrained $\tau$ to force the subgraphs in a cluster to be similar to each other. To gauge the importance of each subgraph concept, we compute the average importance score across all subgraph patterns within the same cluster.

Table 1: Results averaged over ten runs with the standard deviations reported.

| Datasets | Fidelity | | Infidelity | |
|---|---|---|---|---|
| | Train | Test | Train | Test |
| BA-2Motifs | 0.99±0.02 | 0.98±0.02 | 0.52±0.00 | 0.50±0.00 |
| BAMultiShapes | 0.97±0.05 | 0.94±0.07 | 0.50±0.03 | 0.47±0.03 |
| Mutagenicity | 0.87±0.03 | 0.83±0.04 | 0.52±0.03 | 0.49±0.05 |
| NCI1 | 0.58±0.07 | 0.61±0.08 | 0.47±0.05 | 0.48±0.03 |

**Intersections of the subgraph patterns across different clusters.** Now that we have obtained $k$ subgraph patterns, the next step is to remove the redundant ones. For example, if there are two subgraph explanations with similar confidence scores, and one is a subgraph of the other, we should retain only the smaller one. To facilitate this, we engage in subgraph matching on the $k$ subgraph patterns. In cases where two subgraph patterns from the prior step share a common subgraph and their score difference is less than $\epsilon$, we accept the common subgraph while simultaneously removing the two original subgraph patterns from the global explanations. The score assigned to the common subgraph is the maximum score between the two original patterns. This process ensures a more concise and informative set of subgraph explanations.

**Utilize learned global subgraph patterns to explain new instances.** Since we have memorized which subtree patterns that each subgraph pattern is contructed from, when new data instance comes in, we can do something similar to the subtree lookups as described in Section 4.1. First, we look up each subtree embedding of the new instance in the memorized subtrees that consist the subgraph explanations. If any of these subtrees are found within the learned concepts, it signifies that we can utilize the acquired knowledge to provide explanations for the new data instances.

## 5 EXPERIMENTS

### 5.1 DATASETS

In this paper, similar to prior works (Azzolin et al., 2023; Xuanyuan et al., 2023), we focus on graph classifications and conduct experiments on two synthetic datasets BA-2Motifs (Luo et al., 2020) and BAMultiShapes (Azzolin et al., 2023), as well as two real-world datasets Mutagenicity Kazius et al. (2005) and NCI1 (Wale et al., 2008; Pires et al., 2015) on Graph Isomorphism Networks (Xu et al., 2019) to demonstrate the efficacy of our approach. The statistics of these datasets can be found in Appendix A.1.

The BA-2Motifs dataset employs Barabasi-Albert (BA) graphs as base graphs. In this dataset, Class 0 graphs are augmented with five-node cycle motifs, while Class 1 graphs are enriched with "house" motifs. The GNNs are desired to correctly identify the five-node cycle motif as a positive pattern for Class 0 and a negative pattern for Class 1, while recognizing the "house" motif as a positive pattern for Class 1 and a negative pattern for Class 0.

The BAMultiShapes dataset consists of 1,000 Barabasi-Albert (BA) graphs, each containing randomly positioned house, grid, and wheel motifs. Class 0 includes plain BA graphs and those with individual motifs or a combination of all three. In contrast, Class 1 comprises BA graphs enriched with any two of the three motifs. Notably, all motifs, including BA, house, grid, and wheel, are present in both classes, posing a challenge for differentiation. This task underscores the role of global explainers in confirming whether GNNs align with human expectations when known rules exist.

Mutagenicity and NCI1 are real-world chemical and medical datasets, which are challenging for both classification or explainability due to their complex graph structures. Mutagenicity comprises 4,337 molecule graphs, categorized into two classes based on their mutagenic effects. Graphs in Class 0 are mutagenic molecules, and graphs in Class 1 are non-mutagenic molecules. NCI1 contains a few thousand chemical compounds screened for activity against non-small cell lung cancer and ovarian cancer cell lines. The intricacy of these datasets makes it challenging to derive definitive classification rules, even for human experts. This emphasizes the importance of global explainers in facilitating knowledge and pattern discovery within real-world graph data.

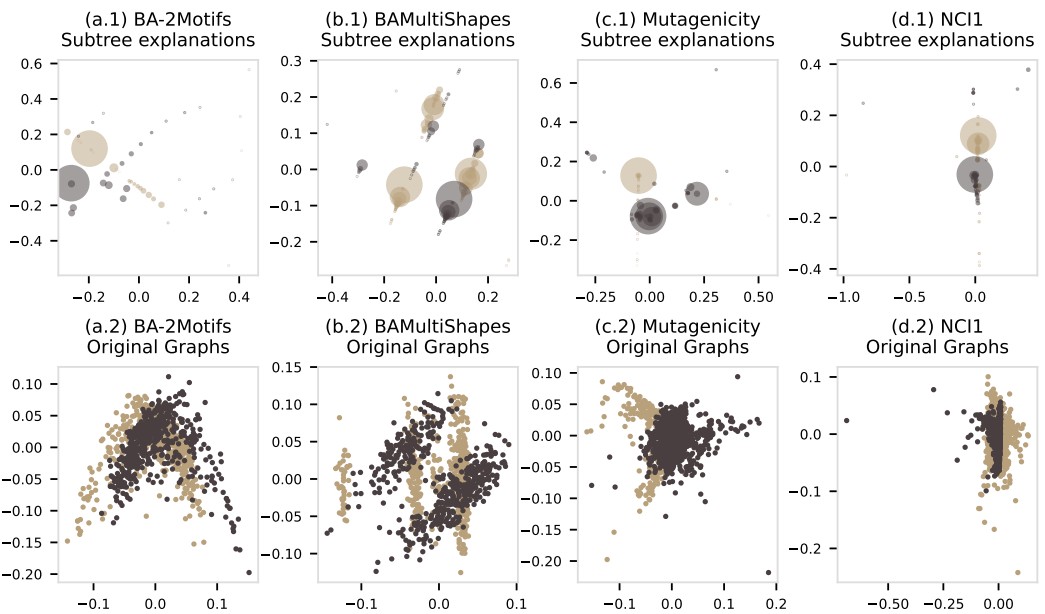

Figure 3: Scatter plots of the global subtree explanations and the original graphs. In the plots of the global subtree explanations, the radius of each marker indicates the number of data samples each subtree covers. The larger a marker is, the more data samples it covers. While the scatter plots of the original graphs in different classes are largely overlapped, the plots of the subtrees in various classes are distanced from each other, clearly representing different concepts.

## 5.2 EVALUATION METRICS

As discussed in Section 1, global explanations are expected to encompass the majority of the data samples while offering meaningful patterns to distinguish between various classes of data. To highlight the robustness of our proposed method in achieving these objectives, we qualitatively evaluate STExplainer based on the following metrics: i) *Fidelity*, which represents the percentage of prediction matching with the original graphs when only the explanation patterns are used to make prediction, for the instances covered by the global explanations. ii) *Infidelity*, which represents the percentage of prediction changes when the explanation patterns are removed relative to the original graphs, for the instances covered by the global explanations. Fidelity and Infidelity are two metrics that have been widely used in the vision domain (Yeh et al., 2019; Zhou et al., 2021). We adopt their definitions of these metrics for graphs to test the faithfulness of the extracted global explanations, which are also known as $(1 - Fidelity-^{acc})$ and $Fidelity+^{acc}$ in Yuan et al. (2022). It is important to note that the subgraph explanations derived from our approach are not prototypes of clusters. Instead, they are pure concepts obtained through subgraph matching. Therefore, we do not measure the purity of these explanations.

## 5.3 RESULTS

We illustrate the global explanations by our approach, and two existing global GNN explainers in Figure 2. Since GCExplainer requires humans in the loop, we did not compare with it in our paper. As shown in Figure 2, GLGExplainer has limitations in delivering clear global explanations. As we discussed in Section 2, the prototypes it generates are latent vectors, lacking clear motifs to represent each prototype. Instead, it provides random local explanations for instances within the cluster. This means that, from the perspective of providing intuitive global explanations, GLGExplainer's outputs remain implicit and require human experts to interpret and draw meaningful conclusions. On the other hand, GCNeuron provides global explanations in the form of logical rules with human-defined premises. However, without prior knowledge, it becomes challenging to define meaningful graph patterns as premises when dealing with the BA-2Motifs dataset. Consequently, the explanations rely on more abstract concepts like the "degree of nodes" or "degree of neighboring nodes", which can make them quite perplexing and challenging for humans to understand. When applied to the Mutagenicity dataset, GCNeuron manually defines 44 premises, including terms like "NO2", "NH2", "NO", "is(C)", "neighbour of C", "2-hop from C". However, GCNeuron fails to recognize "NO2" as a Class 0 motif, even though it's known to be relevant to mutagenic effects Luo et al. (2020).

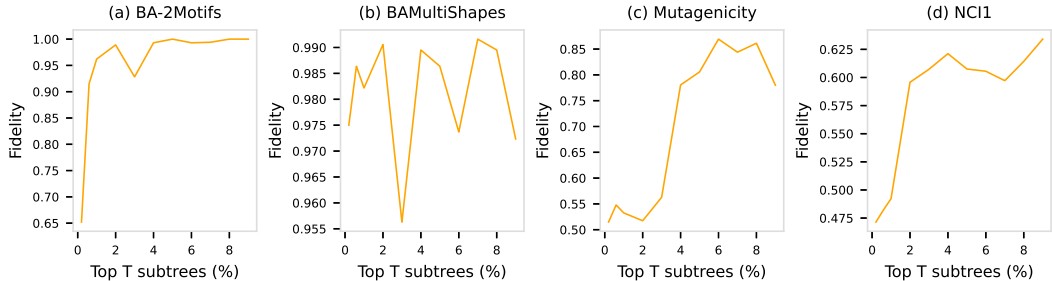

Figure 4: Fidelity of covered data samples relative to the top-T Subtrees (%) as candidate subtrees.

Conversely, STExplainer is able to accurately extract the significant global patterns on BA-2Motifs mentioned in Section 5.1. That is, the five-node cycle motif is the positive graph pattern of Class 0, but the negative graph pattern of Class 1; the "house" motif is the positive graph pattern of Class 1, but the negative graph pattern of Class 0. On the Mutagenicity dataset, STExplainer successfully identifies the "-NO2" and "-NH2" chemical groups as Class 0 patterns with high confidence scores, which are well-known to be associated with mutagenicity, as discussed in previous studies (Ying et al., 2019; Luo et al., 2020; Debnath et al., 1991). Additionally, it identifies "-N2O", "-OCH3" as the Class 0 motifs, and "-CH2", "-OH" as the Class 1 motifs, albeit with relatively lower but still positive confidence. These chemical groups have been widely studied in terms of their mutagenic effects (Hill et al., 1998; Baden & Kundomal, 1987). Explicitly highlighting these chemical groups provides a more comprehensive understanding of how GNNs make decisions and can be valuable for debugging GNNs. Due to the space limit, we have moved the qualitative results on BAMultiShapes and NCI1 to appendices. Please see Appendix A.4 for more results and discussions on these datasets.

We present the performance of our approach on Fidelity and Infidelity metrics in Table 1. These results are obtained by adjusting hyperparameters $T$ and $\lambda$ to fully cover the data samples in the dataset. The high fidelity scores indicate that STExplainer successfully extracts global explanations that closely align with the original GNNs' behavior. The infidelity results, which hover around 0.5, are reasonable for binary classification tasks. This is because, in cases where one class is biased, even if we remove the motifs important to that class, due to the absence of motifs crucial to the other class, the predictions would remain unchanged, resulting in an infidelity score around 0.5. Discussions on the comparison with other global explainers over qualitative evaluations can be found in the appendix.

We additionally illustrate STExplainer is capable to extract subtrees that clearly distinguish between the classes by scatter plots as shown in Figure 3. We obtain the scatter plots via the linear Principal Component Analysis (PCA) (Halko et al., 2011). In the plots of subtree explanations, the larger a marker is, the more data samples the corresponding subtree covers. We can observe that the original data of different classes are largely overlapped. In contrast, the subtrees of various classes are distanced from each other, representing different concepts. For instance, the three lighter clusters in Figure 3 (b.1) represent house, wheel and grid motifs respectively, and the darker clusters are various sub-patterns of BA. For details on these global explanations please also refer to Appendix A.4 and A.3. Furthermore, we report the fidelity performance with respect to $T$ in Figure 4, where we select top $T$ candidate subtrees for training. The results on all the datasets coverage as $T$ increases. In particular, STExplainer achieves a high fidelity on BAMultiShapes with a very small portion of subtrees, where $T = 0.2\%$. It maintains a near optimal fidelity on BA-2Motifs with less than top 1% subtrees selected as the candidates. On real world dataset such as Mutagenicity and NCI1, which contains more complicated sub-structures, it requires $4\% \sim 6\%$ subtrees to converge. Due to space limitations, we have moved the discussions on hyperparameter selection and implementation details to the appendices.

## 6    CONCLUSION

In this paper, we primarily focus on global-level explanations for Graph Neural Networks (GNNs), and introduce a novel approach STExplainer, which offers global insights of the GNN models with the intuitive subtree patterns and subgraph patterns. Our approach inherently allows rapid lookups on the learned concepts while doing inference. Through detailed empirical analysis, we show that the global explanations extracted by STExplainer are more intuitive and human-understandable compared with existing global-level GNN explainers while maintaining the faithfulness and discriminability.

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

# A  APPENDIX

## A.1  DATASET STATISTICS

Table 2: Statistics of datasets.

| Datasets | BA-2Motifs | | BAMultiShapes | | Mutagenicity | | NCI1 | |
|---|---|---|---|---|---|---|---|---|
| | #nodes | #edges | #nodes | #edges | #nodes | #edges | #nodes | #edges |
| mean | 25 | 51 | 40 | 87.5 | 30.3 | 61.5 | 29.9 | 64.6 |
| std | 0 | 1 | 0 | 7.2 | 20.1 | 33.6 | 13.6 | 29.9 |
| min | 25 | 49 | 40 | 78 | 4 | 6 | 3 | 4 |
| quantile25 | 25 | 50 | 40 | 78 | 19 | 38 | 21 | 46 |
| median | 25 | 50 | 40 | 90 | 27 | 56 | 27 | 58 |
| quantile75 | 25 | 52 | 40 | 92 | 35 | 76 | 35 | 76 |
| max | 25 | 52 | 40 | 100 | 417 | 224 | 111 | 238 |
| #graphs | 1000 | | 1000 | | 4337 | | 4110 | |

## A.2  GNN IMPLEMENTATION DETAILS

Table 3: Details of the GNN models used to produce our experimental results, where "hidden" is the latent dimension size, and L is the number of GNN layers.

| Datasets | BA-2Motifs | BAMultiShapes | Mutagenicity | NCI1 |
|---|---|---|---|---|
| L | 3 | 3 | 3 | 3 |
| hidden | 32 | 20 | 64 | 64 |
| pooling | mean | mean | mean | mean |
| layer type | GIN | GIN | GIN | GIN |
| learning rate | 0.01 | 0.01 | 0.01 | 0.01 |
| batch size | 256 | 256 | 256 | 256 |
| epochs | 200 | 200 | 200 | 200 |
| train acc | 1.00 | 0.95 | 0.91 | 0.95 |
| test acc | 1.00 | 0.97 | 0.81 | 0.80 |

## A.3  VISUALIZATION OF GLOBAL EXPLANATIONS ON DATA INSTANCES

Figure 5, 6, 7 visualize the global explanations extracted using our approach on the actual data instances. We can easily observe that the five-node cycles and house motifs are accurately highlighted on the graphs at the corresponding classes in the BA-2Motifs dataset. For BAMultiShapes, the BA patterns for Class 0, as well as house, wheel, grid motifs for Class 1 are clearly illustrated. On Mutagenicity and NCI1, STExplainer is able to highlight functional groups such as "-NO2", "-NH2", "-NO".

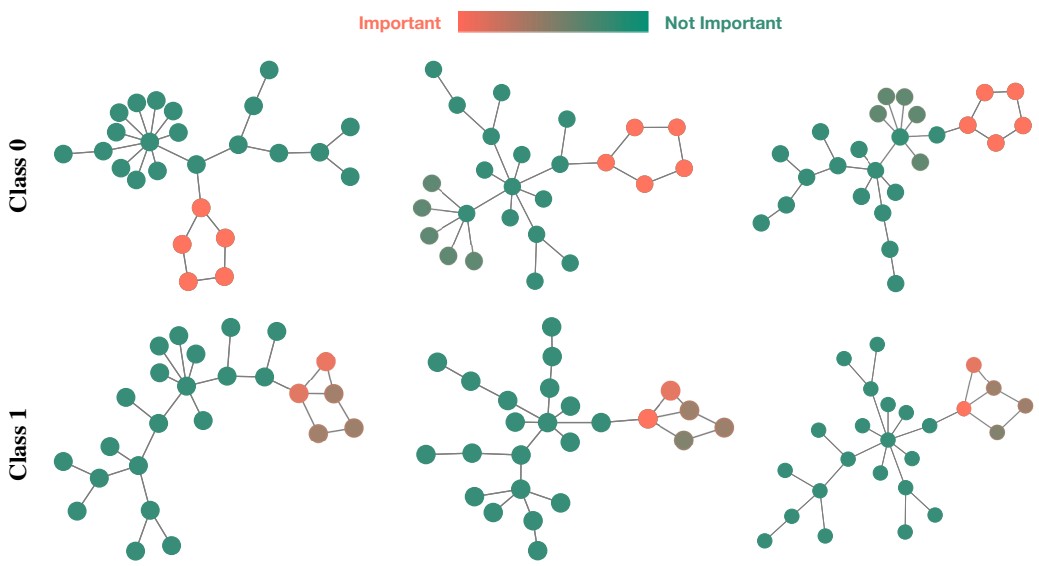

Figure 5: Visualization of the global explanations extracted by STExplainer on the actual data instances of the BA-2Motifs dataset.

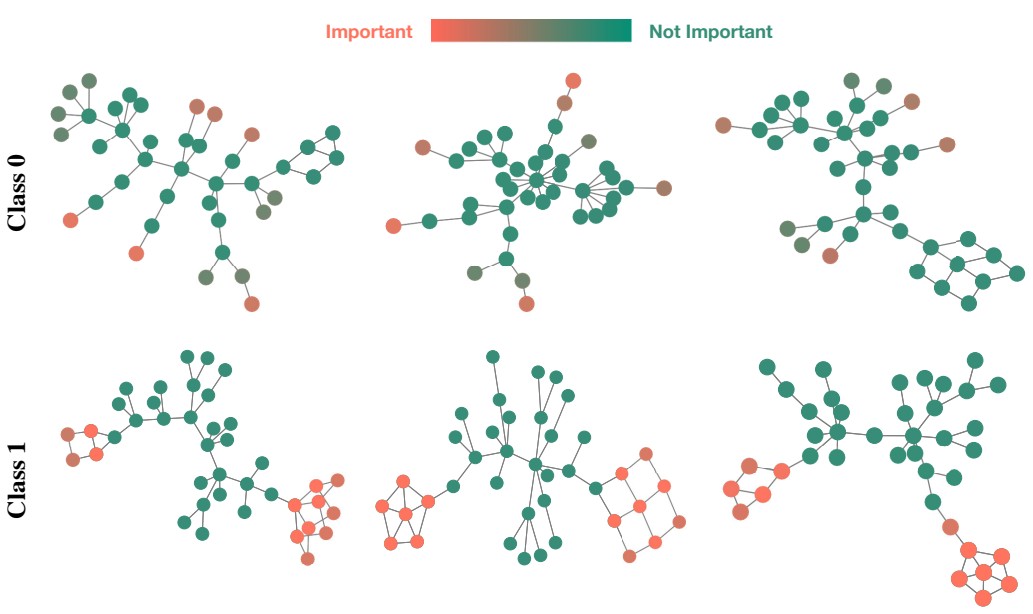

Figure 6: Visualization of the global explanations extracted by STExplainer on the actual data instances of the BAMultiShapes dataset.

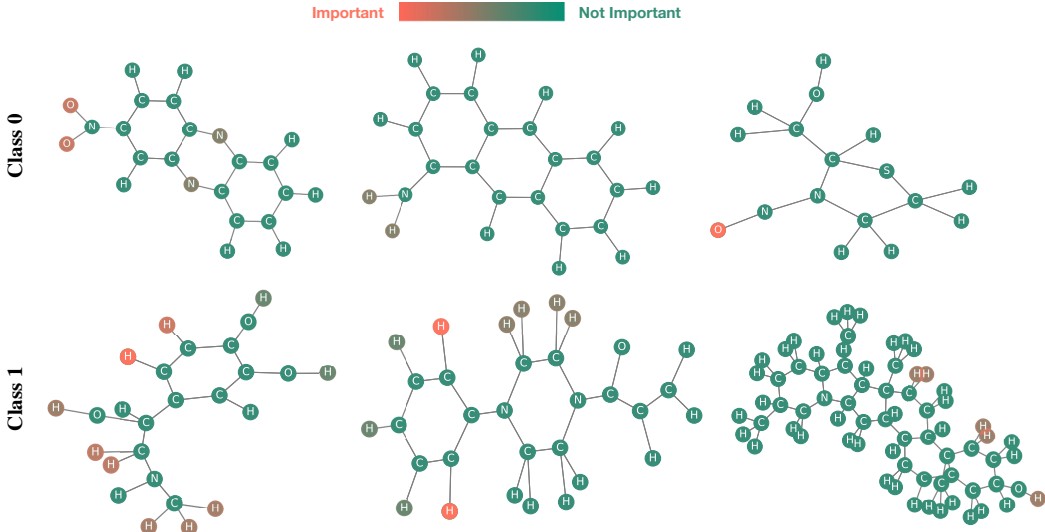

Figure 7: Visualization of the global explanations extracted by STExplainer on the actual data instances of the Mutagenicity dataset.

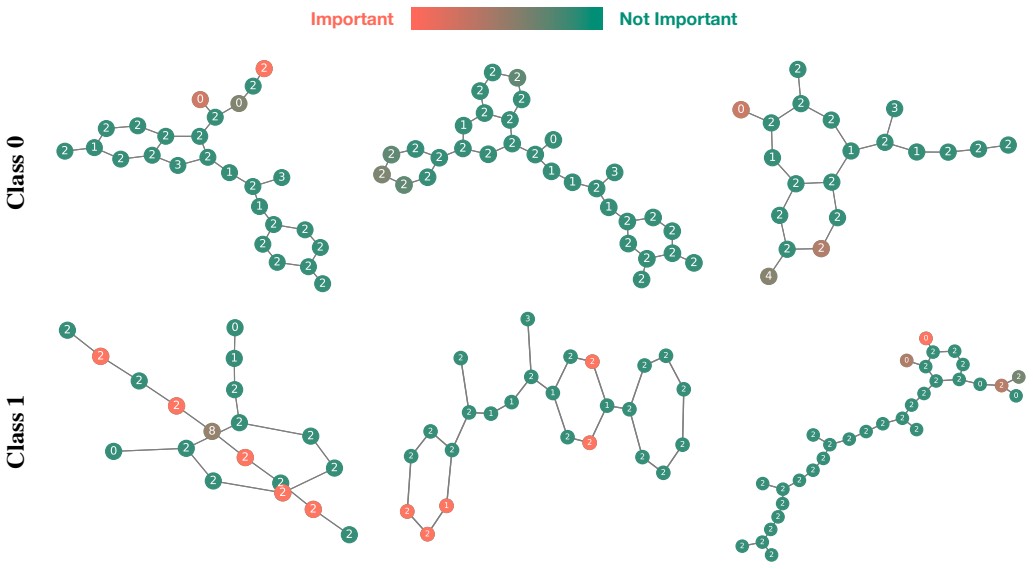

Figure 8: Visualization of the global explanations extracted by STExplainer on the actual data instances of the NCI1 dataset.

## A.4 ADDITIONAL QUALITATIVE RESULTS

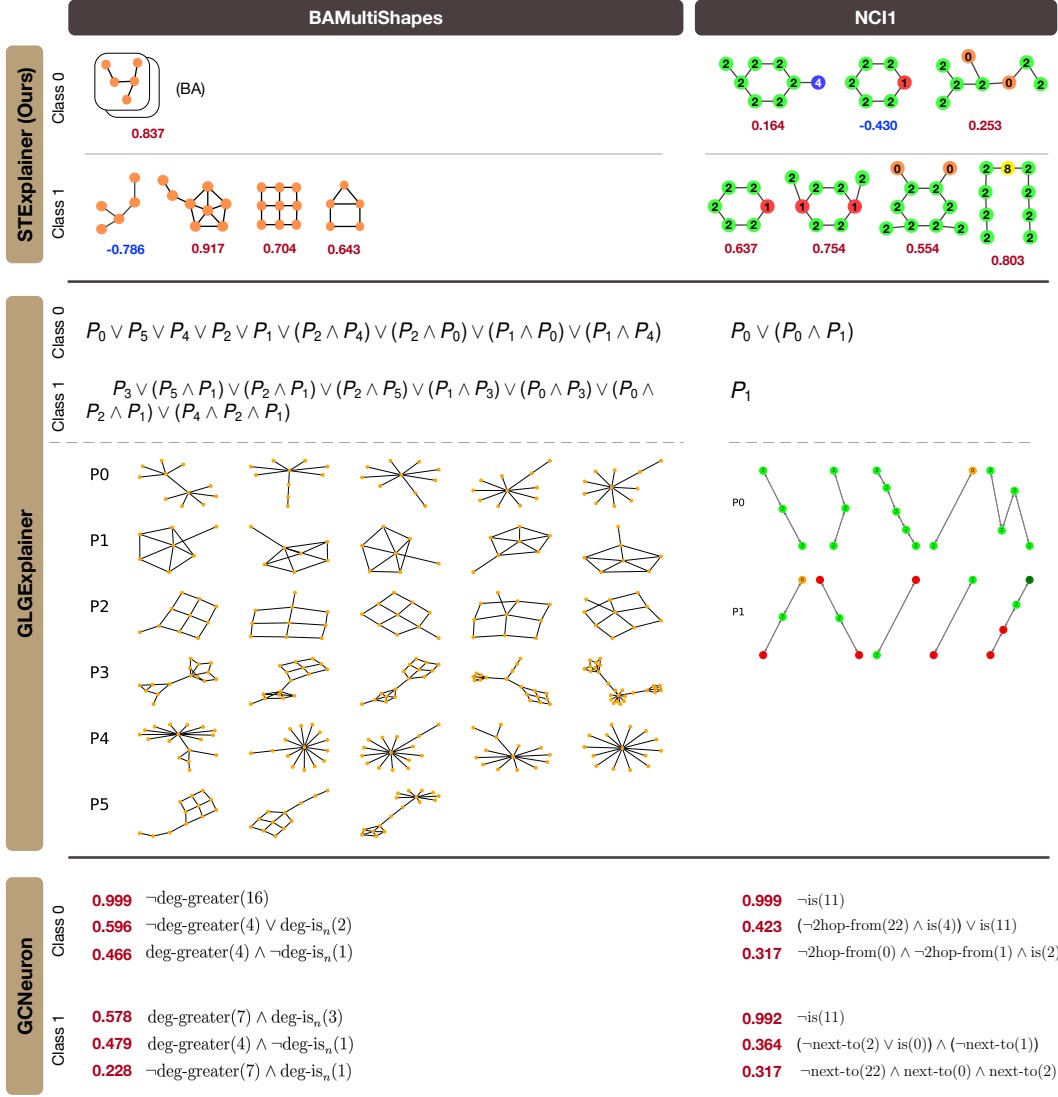

Figure 9: Global explanations by STExplainer (ours), GCNeuron and GLGExplainer on BAMulti-Shapes and NCI1 datasets. We run both baseline methods so that they explain the same GNN models as our approach.

Figure 9 presents the global explanations produced by various global explainers on BAMultiShapes and NCI1 datasets. For the NCI1 dataset, we cannot map the node type numbers to the actual atoms because that information was not available. So, the explanations we provide only include the node type numbers. GLGExplainer generates long Boolean formulas on BAMultiShapes dataset. However, it fails to identify the house motif. Moreover, the predicates in the Class 1 formula, namely $(P_1 \wedge P_3)$, $(P_2 \wedge P_5)$ and $(P_5 \wedge P_1)$, are not faithful to the ground-truth, as they require the presence of multiple grids or multiple wheels in Class 1 graphs. Recall that the ground-truth rules of BAMultiShapes is that Class 0 includes plain BA graphs and those with individual motifs or a combination of all three, whereas Class 1 comprises BA graphs enriched with any two of the three motifs. Furthermore, the Boolean formulas from GLGExplainer on NCI1 is a bit confusing, since $P_0 \vee (P_0 \wedge P_1)$ is logically equivalent to $P_0$. Consequently, for the NCI1 dataset, GLGExplainer only provides random local explanations of each prototype, where Prototype 0 stands for Class 0 and Prototype 1 stands for Class 1. The insights provided by these random local explanations are less

informative. On the other hand, the global explanations from GCNeuron are relatively less intuitive as negations are frequently involved, and they are challenging for humans to understand.

In contrast, our approach successfully identifies all the outstanding motifs for the BAMultiShapes dataset, namely, the house, wheel, as well as grid motifs. In particular, STExplainer recognize the patterns in Barabasi-Albert (BA) graphs as the Class 0 motifs, and house, wheel, grid as the Class 1 motifs. This is reasonable because, as shown in Table 2, all the data samples in BAMultiShapes contain 40 nodes. Hence, if more house, wheel or grid motifs are included in a graph, then less BA patterns will be in it. Given that all the Class 1 graphs contain two out of three motifs in house, wheel or grid, whereas most Class 0 graphs contain at most one of the three motifs, it is reasonable for the GNNs to consider that the Class 0 graphs contains a larger portion of BA patterns than the Class 1 graphs. Additionally, neither STExplainer nor GLGExplainer is able to capture the ground truth rule that graphs contain all of the three motifs are Class 0 graphs, which is as expected, since as shown in Table A.2, the GIN does not achieve perfect accuracy on BAMultiShapes, and there are only around $4\%$ of the graphs contain all of the three motifs. These experimental results have demonstrate that our approach has the potential to provide insights into some occasionally incorrect rules learned by the model. For the NCI1 dataset, the inherent design of STExplainer allows it to capture larger graph patterns than GLGExplainer and GCNeuron.

### A.5    DISCUSSION ON THE COMPARISON WITH GLGEXPLAINER ON FIDELITY

We did not compare with GLGExplainer on the fidelity metric for the following reasons. First, we use a different definition of fidelity from GLGExplainer. GLGExplainer uses "fidelity" to gauge the alignment between the predictions of their E-LEN model and the original GNN, a metric tailored to their specific model and not widely recognized within the broader domain of explainability in neural networks. On the contrary, we adopt the widely accepted concept of fidelity within the field of neural network explainability as we discussed in Section 5, and measure the accuracy by comparing predictions derived solely from the explanations with the original predictions. Second, another metric employed by GLGExplainer, namely "accuracy", bears similarity to the definition of "fidelity" as delineated in our paper. Nonetheless, as expounded upon in Section 5, our reported results stem from hyperparameter tuning efforts aimed at ensuring comprehensive coverage of data samples by the global explanations. In contrast, the design of GLGExplainer presents challenges in achieving full coverage. Specifically, GLGExplainer resorts to manual exclusion of data samples in cases where local explanations fail to meet requirements. For instance, in their implementation on the Mutagenicity dataset, they commenced with 3469 training samples but utilized only 2329 of them for model training. And the accuracy evaluations were conducted exclusively on the subset of data instances that survived this filtering process. These are the reasons that we did not compare with GLGExplainer on the fidelity metrics in our experiments.

### A.6    HYPERPARAMETER SETTINGS

In our experiments, we use the following hyperparameter settings. For BA-2Motifs and BAMultiShapes, we choose $T = 2\%$, $M$ is determined by thresholding the subtrees of absolute scores exceeding $0.03$. For Mutagenicity, we choose $T = 4\%$, $M$ is determined by thresholding the subtrees whose absolute scores exceeding $0.01$. For NCI1, we choose $T = 5\%$, $M$ is determined by thresholding the subtrees whose absolute scores exceeding $0.001$. The learning rate to train the models for all the datasets in Step 2 of Figure 1 is $lr = 0.002$, and the number of epoch=2000. For all the datasets, we set $\lambda = 0.1, \tau = 2.00$ and allow a maximum of $k_{max} = 40$ clusters.

Table 4: Exact number of clusters.

| Datasets | BA-2Motifs | BAMultiShapes | Mutagenicity | NCI1 |
|---|---|---|---|---|
| $k$ | 10 | 21 | 32 | 36 |

Table 4 shows the exact number of clusters, i.e., $k$ that is used in the experiments. Since the required number of clusters does not go beyond 40, we set $k_{max} = 40$ to allow the cluster algorithm to determine the number of clusters required for each dataset.

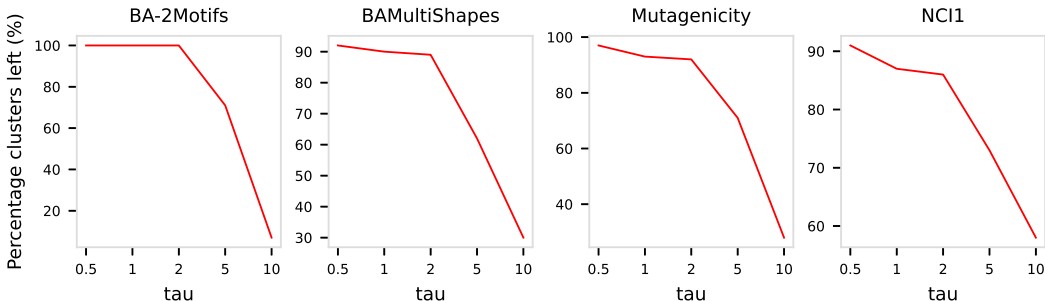

Figure 10: Percentage of valid clusters (not discarded) over ten runs with respect to various choice of $\tau$.

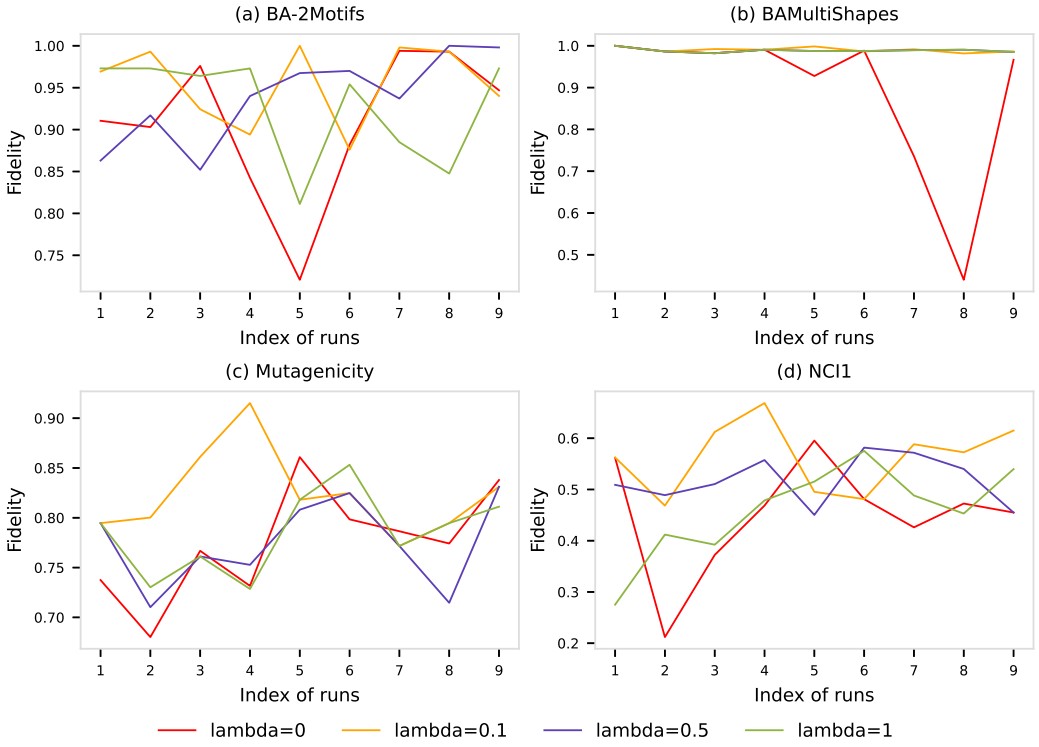

Figure 11: Fidelity performance with respect to $\lambda$

