# OpenReview forum: "STExplainer: Global Explainability of GNNs via Frequent SubTree Mining"
_ICLR.cc/2024/Conference — Submitted to ICLR 2024_

### Official Review · Reviewer_zuiK · 2023-10-13

**Soundness:** 1 poor
**Presentation:** 2 fair
**Contribution:** 2 fair
**Rating:** 3
**Confidence:** 5

**Summary:**

The study introduces STExplainer, a global GNN explanation mechanism leveraging frequent subtree mining. Initially, the method isolates the Top-K frequent L-hop subtrees, employing the root node's embedding as the subtree's representation. Subsequently, it computes scores for each subtree and selects the top M as the global subgraph explanation. Additionally, the method incorporates overlapping graph combination techniques and a subgraph matching algorithm to pinpoint intricate and basic subgraphs.

**Strengths:**

1. The global-level GNN explainer is an important topic that needs further research.
2. Using subtree to construct explanations is interesting.
3. The paper is well-organized and easy to follow.

**Weaknesses:**

1. My first concern regarding the proposed method is its complexity. After selecting the Top-K subtrees, the method merges overlapping subtrees to form subgraph patterns, necessitating access to all graph instances. Furthermore, the technique involves extracting intersection subgraphs from these patterns within a cluster through subgraph matching and employs additional subgraph matching to eliminate redundant subgraphs. Repeatedly accessing all graph instances can amplify the method's complexity. It is imperative for the authors to delve into the method's computational complexity and provide experimental comparisons with other baseline approaches.
2. An initial subtree candidate is derived from an L-hop subgraph, and the size of the subtree candidate will significantly influence the results. This implies that the method's efficacy hinges on the configuration of the target GNNs. I highly recommend authors explain how L affects the method and use an experiment to show it.
3. The suggested approach employs a Multi-Layer Perceptron (MLP) to obtain a node embedding matrix from the subtree feature matrix. This embedding matrix subsequently represents the score vector for all subtrees. However, integrating an additional neural network as a scoring mechanism complicates interpretability. Also, as the score of an individual subtree contributes to determining the score for the overall subgraph explanation and the overlapping of subgraph patterns, the construction of the initial subtree will significantly impact the results.
4. Subtrees are ranked according to their frequency. However, as depicted in Figure 4, the performance on the NCI datasets appears to be closely linked to some infrequent subtrees. Does this suggest that frequency might not be an optimal criterion for subtree selection?
5. In the process of selecting the Top-K subtrees, there is no provision to detect isomorphic subtrees. As a result, isomorphic subtrees might be assigned differing importance scores.

**Questions:**

Please refer to Weaknesses section.

---

> ### Author Response · Authors · 2023-11-23
>
> We would like to express our concern regarding the validity of this review, as the weaknesses raised by the reviewer appear to primarily stem from misunderstandings. To clarify these points, we provide an explanation of our design in response to each of the mentioned weaknesses.
>
> **W1**: There is a misunderstanding here. After selecting the Top-M subtrees, our method exclusively works with these M subtrees, eliminating the need to access all graph instances repeatedly. The information about whether two subtrees are in the same data instance is obtained during the initial access to the data instances. Therefore, there is no redundant access to all graph instances, showcasing the efficiency of our design. This highlights that the method's complexity is not amplified, reinforcing the effectiveness of our approach in utilizing subtrees instead of subgraphs or data instances.
>
> **W2**: We would like to clarify a potential misunderstanding. $L$ is NOT a hyperparameter of our approach but represents the number of GNN layers, as mentioned in our paper. That is, when dealing with a pretrained $L$-layer GNN, we need to consider $L$-hop subtrees, where $L$ is fixed and is the same as the number of layers in the GNN in question. Altering $L$ can impact the performance of the original GNN in question. Therefore, changing $L$ wouldn't lead to a fair experiment for evaluating our explainer's performance.
>
> **W3**: The main goal of our paper is to provide understandable global-level GNN explanations for humans, not to tackle the broader challenge of neural network interpretability. We use the MLP to learn the importance of candidate subtrees constructed based on their frequency, and this importance helps explain the GNNs. It's important to note that MLP is not mandatory; we could learn the importance of subtrees as learnable weights. We used an MLP because it allows us to potentially consider more detailed information, such as subtree embeddings, while learning the scores. Additionally, many recent works in local-level GNN explainability, a parallel research domain, use MLPs for generating explanations on data instances [1,2,3]. Therefore, we believe that using an MLP to learn scores is not a major issue.
>
> [1] Mohit Bajaj, Lingyang Chu, Zi Yu Xue, Jian Pei, Lanjun Wang, Peter Cho-Ho Lam, and Yong Zhang. Robust counterfactual explanations on graph neural networks. Advances in Neural Information Processing Systems, 34:5644–5655, 2021.
>
> [2] Dongsheng Luo, Wei Cheng, Dongkuan Xu, Wenchao Yu, Bo Zong, Haifeng Chen, and Xiang Zhang. Parameterized explainer for graph neural network. Advances in neural information processing systems, 33:19620–19631, 2020.
>
> [3] Yaochen Xie, Sumeet Katariya, Xianfeng Tang, Edward Huang, Nikhil Rao, Karthik Subbian, and Shuiwang Ji. Task-agnostic graph explanations. Advances in Neural Information Processing
> Systems, 35:12027–12039, 2022.
>
> **W4**: The reviewer's interpretation for Figure 4 might be misleading. First, the figure only displays results for the top frequent subtrees. We do not have information on any “infrequent” subtrees. Only the results for top frequent subtrees are shown. Second, as we discussed in Section 5, Figure 4 illustrates that, despite some variance across runs, the results on all the datasets coverage as T increases. The variance stems from rerunning the model for different runs. Importantly, this figure and the corresponding experiments aim to choose the hyperparameter T on various datasets. It demonstrates that BA-2motifs and BAMultiShapes datasets require fewer subtrees compared to Mutagenicity and NCI1, aligning with our expectations as real-world datasets are more complex and contain more substructure types.
>
> **W5**: Isomorphic subtrees will not have differing importance scores because we employ GNNs to identify subtrees, rather than manually selecting them. GNNs will always ensure that isomorphic subtrees yield identical embeddings. Consequently, the scores computed from these identical subtree embeddings would be the same.
>
> We hope our explanations resolve your misunderstandings toward this work. We would appreciate it if the reviewer can reassess the quality of this work based on the research question it is addressing, its technical novelty as well as the outstanding performance in comparison to SOTA methods.

---

### Official Review · Reviewer_tXc8 · 2023-10-20

**Soundness:** 2 fair
**Presentation:** 3 good
**Contribution:** 3 good
**Rating:** 5
**Confidence:** 5

**Summary:**

The paper introduces a new approach to global-level explainability in GNNs called "SubTree Explainer." This method focuses on mining important rooted subtrees across a dataset, offering a more comprehensive view of GNN behavior compared to existing local-level methods.

**Strengths:**

1. The authors provide a more intuitive and validated method for GNN explainability.
2. STExplainer directly mines important rooted subtrees across the entire dataset, making the process more efficient and focused.
This paper is well-articulated with a clearly defined objective.
3. This paper demonstrates the effectiveness of STExplainer in generating high-quality  global explanations on synthetic and real-world datasetss.

**Weaknesses:**

1. The experimental section could be more detailed.

**Questions:**

1. It is expected that the authors could explain the reasons forwhy not select accuracy as one of the metrics?
2. In Table 1, the authors present the results of their model,  suggest that they could choose some baselines for comparison to better illustrate the model's effectiveness.

---

> ### Author Response · Authors · 2023-11-23
>
> **W1**: Due to space limitations, we had moved more details about the experiments to Appendix A.1~A.6.
>
> **Q1**: We have evaluated the Fidelity in accuracy in Section 5. As we defined in Section 5.2. The Fidelity metric in our paper, represents the percentage of prediction matching with the original graphs when only the explanation patterns are used to make prediction, for the instances covered by the global explanations. This metric is called “accuracy” in some literature. But to be consistent with the common terminology in this domain [1], we called it “Fidelity” in our paper.
>
> [1] Hao Yuan, Haiyang Yu, Shurui Gui, and Shuiwang Ji. Explainability in graph neural networks:
> A taxonomic survey. IEEE transactions on pattern analysis and machine intelligence, 45(5):
> 5782–5799, 2022.
>
> **Q2**: The main focus of our approach is to provide higher quality global-level explanations for GNNs. That is, we provide global explanations in a different form from the previous approaches, where our explanations are more intuitive and human-understandable graph concepts, and can potentially be easier to use in the downstream analysis (e.g. constructing building blocks), while the existing approaches provide either the latent representations or the human-defined rules as explanations (see Figure 2 and Figure 9). The empirical results demonstrate that our method provides significantly higher quality global explanations than the existing methods as also mentioned by Reviewer GgEn. Moreover, an advantage of our method is that we can quantitatively evaluate it by removing subgraphs from the original graph data. In contrast, existing methods provide either latent representations or human-defined rules, making such removal infeasible. As a result, these existing methods didn't conduct quantitative comparisons with any baseline in their own paper. Although we couldn't provide a quantitative comparison with them, we quantitatively assessed our approach using widely accepted metrics like Fidelity and Infidelity, as shown in Table 1. The high quantitative performance indicates that STExplainer effectively extracts global explanations aligning closely with the original GNNs’ behavior.

---

### Official Review · Reviewer_GgEn · 2023-10-22

**Soundness:** 2 fair
**Presentation:** 2 fair
**Contribution:** 2 fair
**Rating:** 5
**Confidence:** 3

**Summary:**

The authors introduce a novel method to extract global explanations for a GNN via rooted sub-trees on a dataset. The authors method works by enumerating all possible L-hop subtrees which is more efficient than enumerating all possible subgraphs.  They take the top T subtrees that belong predominantly to a single target class (ensuring trees that belong to multiple classes are ignored). From these trees they obtain weighted embeddings. The weighted embedding is passed into the classifier of the original GNN resulting in the final prediction values of each output class prior to softmax layers. They choose the M most important subtrees that they obtain by minimizing a loss function that tries to find the most important subtrees for a target class while penalizing embeddings with larger weights. From these subtrees they combine overlapping ones into subgraphs. They cluster them via k-means in the embedding space for these subgraphs and obtain a representative subgraph per cluster by subgraph matching S subgraphs from the cluster. These subgraphs are the global explanations for a class.

The authors then conduct experiments on 2 synthetic datasets (BA2-Motifs and BAMultiShapes)  and 2 real datasets ( Mutagenicity and NCI1) and evaluate the fidelity and infidelity on these datasets. The authors also show cases where other Global explainer methods do not provide adequate explainability.; GLGExplainer and GCNeuron make errors on BA-2Motifs dataset and Mutagenicity respectively. They also visually show how the different global explanations distinct from different classes; that is when they are done with their method there explanations are not overlapping.

**Strengths:**

The paper provides a novel approach to a relatively new research direction of global explanations in GNNs. The method is novel and the pipeline to extract explanations makes sense and intuitively seems like a step in the right direction for global explanations of GNNs. Each step of the pipeline also makes sense intuitively to extract explanations that do not overlap, are significant, and is more efficient than enumerating all possible explanations. The experiments also seem initially promising, and there are certain cases where this method does significantly better than existing global GNN explainers. The experiment methodology also lists all applicable hyperparameters to reproduce the experiments.

**Weaknesses:**

Their method they introduce is novel and intuitive but certain choices in their method seem arbitrary and perhaps even sub-optimal without justification. For instance, when conducting k-means clustering they choose k so that the centers of the clusters are distant enough which is parameterized by a hyperparameter tau. Clearly the choice of tau influences the choice of k and hence the rest of the method it is important to justify this choice. In all their experiments they set tau =2 and it is not clear why this particular value. Also as the authors have stated they want to incentivize the center of the clusters to be close to each other however it would be relevant and important to explore the performance of this method with different choices of this hyperparameter and hence k.

Also after extracting overlapping subtrees which leads to subgraphs their method then does k-means clustering on the embeddings. From this cluster they sample S subgraphs and average to obtain a representative subgraph for the cluster. Using this method to obtain a prototype seems arbitrary, why not just use the centroid itself or other prototype learning techniques to obtain the representative sample.

Finally, the experimental section shows promise for this method. However, the experiments are not exhaustive and only work with 2 datasets that are real and 2 that are synthetic. The authors justify why they do not compare to other global explainers however more cases where other methods fail and they succeed would provide a much stronger case for their approach. The authors also use 1 choice of major hyperparameters such as tau, and lambda. They should also show experiments on how the choice of these parameters affect their method.

**Questions:**

If further justification for particular choices used in their method are made. For instance why is averaging S subgraphs from a cluster the optimal choice for a representative subgraph.

Also why choose tau=2 for all experiments, what other good choices are there for this hyperparameter?

The experiments also do not seem sufficient in showing their method’s performance especially with the limited choice of hyperparameters. If further justification to why these experiments are sufficient this would alleviate concerns of the methods feasibility. Alternatively it would be best to show more experiments in a wider range of settings, with greater choices of hyperparameters.

Lastly, there are some typos:

In the second last sentence of the Introduction the authors wrote ‘methos’ instead of ‘methods.’

In the Introduction when listing the authors’ contributions; specifically the last sentence of ii) instead of writing ‘...concepts rather latent representation...’ it should be ‘...concepts rather than latent representation...’

In the appendix specifically section A5 there are several incorrect uses of forward quotations and backwards quotation marks.

Also in the appendix the title of section A6 is mispelled it should be ‘Hyperparameter’

These should be fixed.

---

> ### Author Response · Authors · 2023-11-23
>
> Thank you very much for the detailed review.
>
> **W2&Q1**: As we discussed in the paragraph of “Intersections of the subgraph patterns in the same cluster” in Section 4.2, after we “sample S subgraphs” and before we “average to obtain a representative subgraph for the cluster”, we perform subgraph matching on the S subgraphs, ultimately yielding the intersection of these S subgraphs. The intersection is not necessarily one of the S subgraphs, but likely a new one, hence it can be better represented by the mean of the corresponding S subgraphs in comparison with a prototype in them.
>
> **W3**: We provide global explanations in a different form from the previous approaches, where our explanations are more intuitive and human-understandable graph concepts, while the existing approaches provide either the latent representations of clusters or the human-defined rules as explanations (see Figure 2 and Figure 9). It's not that existing approaches sometimes fail to give intuitive global graph concepts; rather, they cannot provide them at all. We hope our work can help the community to give more attention to the global graph explanations because of their intuitiveness and potential for further analysis (e.g. constructing building blocks). Additionally, the existing works conducted experiments on 3~4 datasets, such as GLGExplainer on 3 datasets and GCNeuron on 4 datasets. Therefore, we believe demonstrating the effectiveness of our approach with 4 datasets is sufficient in comparison.
>
> **W1&Q2&Q3**: We understand the reviewer's concern regarding hyperparameter settings and would like to provide clarification. Regarding $\tau$ and $k$-cluster, we would like to clarify that we did not fix $k=40$, but only let $k_{max}=40$, where $k_{max}$ is an argument to reproduce the results in this paper using our code, meaning that a maximum of 40 clusters is allowed. In our implementation, the exact number of clusters used for each dataset is dynamically determined. We have updated the manuscript by reporting the exact number of $k$ in the appendix (highlighted in red). For $\tau$, typically we want $\tau$ to be small, such that we are able to find the intersections on the S subgraphs in each cluster. As discussed in Section 4, we would discard a cluster in cases no common subgraphs are found in the cluster. We chose $\tau=2$ for the datasets since we find that we only need to discard a small ratio of clusters when $\tau \leq 2$. The choice of $\lambda$ is set to $\lambda=0.1$ as it is relatively more stable in achieving high performance. With other settings of $\lambda$, our method is still able to achieve high performance, which further demonstrates the robustness of our proposed pipeline. We updated the manuscript by the experiments for choosing $\tau$  and $\lambda$ in Appendix A.6. Regarding $T$, our choice is explained in Section 5.3 with Figure 4. The figure illustrates that BA-2motifs and BAMultiShapes datasets require fewer subtrees compared to Mutagenicity and NCI1. This aligns with expectations since real-world datasets are more complex and contain more types of substructures.
>
> **Q4**: Thanks a lot for pointing out the typos, we have updated the manuscript accordingly (highlighted in red).
>
> We hope our explanations and additional experiments resolve your concerns. Your response would be deeply appreciated.

---

### Official Review · Reviewer_zhbU · 2023-10-24

**Soundness:** 3 good
**Presentation:** 3 good
**Contribution:** 3 good
**Rating:** 6
**Confidence:** 3

**Summary:**

This paper proposes a method that provides global explanations for the GNN inferences. Unlike the conventional global explanations that aggregate local explanations, the proposed method utilizes frequent subtrees within the dataset. The weighted sum of the embeddings of the rooted nodes of the subtrees is put into the GNN classifier, and the weights are optimized so that the value of the softmax function for the designated class is maximized. Thus the highest weights are assigned to the globally important subtrees for the class. Moreover, these subtrees are classified into clusters and aggregated into subgraphs according to the overlapped subtrees. The experimental results using two artificial datasets and two real-world chemical datasets show that the proposed method provides human-readable global explanations better than conventional global explanation methods.

**Strengths:**

This paper tackles the difficulties of global explanations for GNN inferences where human-readable global explanations are hard to obtain because of the wide range of possibilities of graph structural features. It gives a solution by utilizing frequent subtrees within the dataset and aggregating them into subgraphs suitable for each instance. It empirically shows that the proposed method can provide easily understandable global explanations for two artificial datasets and two real-world chemical datasets better than conventional global explanation methods for GNNs.

**Weaknesses:**

The novelty of the proposed method is somewhat weak. In fact, it successfully provides human-readable global explanations for the datasets used; however, the method itself is some kind of a surrogation model for the given GNNs and it seems possible to be realized by using several traditional methods such as those using graphlet kernels.

Moreover, the comparison to the conventional methods is shown only by using anecdotal examples such as in Figure 2. More evaluation compared to the conventional methods is required in order for readers to know how the proposed method can be used in their own tasks.

In addition, Figure 3 is difficult for readers to understand because there is not enough explanation, especially for the plots of the original graphs.

**Questions:**

- Is there any quantitative evaluation results in comparison to the conventional methods?

---

> ### Author Response · Authors · 2023-11-23
>
> **W1**: We want to highlight the novelty of our approach in two key aspects. First, our focus is on delivering higher-quality global explanations for GNNs in the form of graph concepts. In this realm, there are only a few existing works, which are GCExplainer, GLGExplainer, and GCNeuron. As detailed in Section 2, these methods aren't surrogate models, and the global explanations they generate are either less intuitive latent clusters or human-defined rules (see Figure 2 and Figure 9). In contrast, our method offers graph concepts as explanations, which are more intuitive and human-understandable, as highlighted by Reviewer GgEn. These concepts are also easier to use for further analysis compared to conventional approaches, such as constructing building blocks. Second, we introduce a novel approach by utilizing subtrees instead of enumerating subgraphs. This significantly reduces search complexity and has been recognized as "novel" and "a step in the right direction for global explanations of GNNs" by Reviewer GgEn. We learn the importance of subtrees across the entire dataset at a high level, without constructing a surrogate model to mimic GNN behavior on local instances. Our empirical results validate the effectiveness of this novel approach. In summary, we propose to extract global explanations with subtrees which are novel to this domain, producing intuitive global graph concepts that existing methods cannot provide.
>
> **W2&Q1**: As we discussed in the response to W1, we provide global explanations in a different form from the previous approaches. We offer more intuitive and human-understandable graph concepts, while existing approaches typically present either latent representations or human-defined rules as explanations (refer to Figure 2 and Figure 9). An advantage of our method is that we can quantitatively evaluate it by removing subgraphs from the original graph data. In contrast, for existing methods, which provide latent representations or human-defined rules, such removal is not feasible, making quantitative comparisons challenging. These existing methods also didn't conduct quantitative comparisons with any baselines. Although we couldn't provide a quantitative comparison, we quantitatively assessed our approach using widely accepted metrics like Fidelity and Infidelity, as shown in Table 1. The high quantitative performance indicates that STExplainer effectively extracts global explanations aligning closely with the original GNNs’ behavior. Beyond the advantage of intuitiveness, STExplainer's ability to extract clear global-level graph concepts opens up possibilities for applications like constructing building blocks in tasks such as graph design. Additionally, the global explanation obtained by our approach can potentially be used to construct supernodes, offering opportunities to enhance network performance.
>
> **W3**: Thank you for the advice. Figure 3 contains scatter plots of the global subtree explanations and the original graphs. In the plots of the global subtree explanations, the radius of each marker indicates the number of data samples each subtree covers. The larger a marker is, the more data samples it covers. While the scatter plots of the original graphs in different classes are largely overlapped, the plots of the subtrees in various classes are distanced from each other, clearly representing different concepts. This demonstrates the significance of our approach in producing clear subtree concepts. We have added this note to the caption of Figure 3 (highlighted in blue).

---

### Meta-Review · Area_Chair_TkB1 · 2023-12-09

**Metareview:**

**Summary**
This paper proposes using frequent subtree mining to extract interpretable explanations from GNNs. Specifically, the proposed method first mines subtree patterns that emerge during message passing between neighbors in GNNs, instead of directly mining subgraphs of given graphs to reduce the cost for enumeration. Subsequently, subgraph explanations are constructed from these mined subtrees. The obtained subgraph explanations has been demonstrated on synthetic and real-world datasets.

**Strengths**
- The idea of using subtree patterns to reduce the computational cost and subsequently constructing subgraph explanations from them is interesting.
- The pipeline of the proposed method is carefully designed.
- The topic explored in this paper, specifically addressing the global explainability of GNNs, is relevant in the community and deserves further exploration.

**Weaknesses**
- The empirical evaluation lacks convincing evidence due to the limited variety of datasets, as highlighted by the reviewers. This limitation is crucial, and the overall quality of the paper could be significantly improved by conducting additional experiments with a more diverse set of datasets.
- Additionally, more careful treatment of hyperparameter evaluation is necessary. While some results have been added in the revised version, they are not comprehensive or thorough.
- I recommend comparing the proposed method to a naïve baseline, such as (significant) subgraph mining, that directly enumerate relevant subgraphs. If there is a substantial difference in the subgraph explanations obtained by the proposal, it could indicate the unique properties of GNNs, which would be an interesting finding.

**Justification For Why Not Higher Score:**

I have carefully reviewed the paper, and the weaknesses of the paper, as outlined above and also highlighted by some of the reviewers, are crucial and must be addressed for the publication of this paper. I appreciate the authors' response and the revision made during the author-reviewer discussion phase, while it requires resubmission in my opinion. Given the potentially interesting technical novelty of this paper, I strongly recommend addressing all the raised issues by the reviewers for substantial improvement before considering resubmission.

**Justification For Why Not Lower Score:**

N/A

---

### Decision · Program_Chairs · 2024-01-16

Reject